# Risk Factors Involved in the High Incidence of Bladder Cancer in an Industrialized Area in North-Eastern Spain: A Case–Control Study

**DOI:** 10.3390/jcm12020728

**Published:** 2023-01-16

**Authors:** José M. Caballero, José M. Gili, Juan C. Pereira, Alba Gomáriz, Carlos Castillo, Montserrat Martín-Baranera

**Affiliations:** 1Department of Urology, Hospital Universitari Mútua Terrassa, Plaza Dr. Robert 5, 08221 Terrassa, Spain; 2Department of Paediatrics, Obstetrics & Gynaecology and Preventive Medicine and Public Health, School of Medicine, Autonomous University of Barcelona, Edificio M Campus Universitario UAB, 08193 Barcelona, Spain; 3Department of Clinical Epidemiology, Consorci Sanitari Integral, Avinguda Josep Molins 29-41, 08906 Hospitalet de Llobregat, Spain

**Keywords:** bladder cancer, risk factors, occupational exposure, smoking, analgesics

## Abstract

Bladder cancer (BC) is the most common of the malignancies affecting the urinary tract. Smoking and exposure to occupational and environmental carcinogens are responsible for most cases. Vallès Occidental is a highly industrialized area in north-eastern Spain with one of the highest incidences of BC in men. We carried out a case–control study in order to identify the specific risk factors involved in this area. Three hundred and six participants were included (153 cases BC and 153 controls matched for age and sex): in each group, 89.5% (*n* = 137) were male and the mean age was 71 years (range 30–91; SD = 10.6). There were no differences between groups in family history, body mass index, or dietary habits. Independent risk factors for CV were smoking (OR 2.08; 95% CI 1.30–3.32; *p* = 0.002), the use of analgesics in nonsmokers (OR 10.00; 95% CI 1.28–78.12; *p* = 0.028), and profession (OR: 8.63; 95% CI 1.04–71.94; *p* = 0.046). The consumption of black and blond tobacco, the use of analgesics in nonsmokers, and occupational exposures are risk factors for the development of BC in this area, despite the reduction in smoking in the population and the extensive measures taken in the last few decades in major industries to prevent exposure to occupational carcinogens.

## 1. Introduction

Bladder cancer (BC) is the most common of those affecting the urinary tract. Worldwide, BC is the seventh most common cancer diagnosed in men and the seventeenth in women. When only developed countries are considered, it ranks fourth and ninth in men and women, respectively. BC represents 4.4% of all new cancer diagnoses (excluding nonmelanoma skin cancer) in the United States and Europe [1].

The incidence of BC in Spain, adjusted for age to the Standard European Population, is 20.08 cases per 100,000 inhabitants (95% CI 13.9–26.3), one of the highest in Europe [2,3]. Vallès Occidental is a region of Catalonia located in the northeast of Spain with an extended industrial tradition, mainly textiles, in which a markedly elevated incidence of BC was detected in men in the 1990s [4]. This trend remains at present [5], with a crude rate of 62.6 (95% CI 55.0–70.1) in men and 6.8 (95% CI 4.4–9.3) in women, and an annual rate adjusted for the standard European population of 85.3 (95% CI 75.0–95.5) in men and 7.0 (95% CI 4.5–9.5) in women. In addition, although we do not have specific data on other types of cancer, the crude rate per 1000 inhabitants of active neoplasia in West Vallès Occidental in 2015 was significantly higher than that of the rest of Catalonia (29.17 vs. 21.85, respectively) [6].

The main risk factor for the development of bladder cancer is smoking, accounting for 50% of the cases [7,8]. The occupational exposure to carcinogens is the second most relevant risk factor, the estimation being that up to 10% of bladder cancers have their origin in occupational exposure [9].

Given the high incidence of BC in Vallès Occidental, the current study aims to identify the independent risk factors that may favour the development of BC in this setting.

## 2. Materials and Methods

A case–control study was designed to assess BC risk factors. Cases were identified in the area of the Hospital Universitari Mútua Terrassa, which serves a population of more than 260,000 inhabitants; inclusion criteria were being aged 18 years or more, and having a histologically confirmed diagnosis of primary BC during the years 2018–2019. Controls were obtained from hospital-recruited individuals without BC, matched with cases for sex and age (±2 years), during the same period. Both the cases and the controls had to be residents of the West Vallès Occidental health area. Cases with nonurothelial bladder tumors and recurrences were excluded from the study.

A sample size of at least 106 cases and 106 controls matched for age and sex was estimated, accepting an alpha risk of 0.05 and a beta risk of 0.2 in a bilateral contrast, to detect a minimum odds ratio of 2.5. It was assumed that the proportion of exposure to any of the studied factors in the control group would be 0.2 [10].

A survey was developed to obtain information through a direct interview, always conducted by the same urologist, who inquired about the patient’s demographic and medical data, as well as about the risk factors under study. Possible risk factors included medical family history of BC; area of habitual residence; consumption of toxic substances, including black and blond tobacco (number of cigarettes per day and years of consumption) and alcohol intake (grams of alcohol per day and years of consumption); characteristics of diet in relation to the consumption of caffeinated or decaffeinated coffee (number of cups of coffee and years of consumption), intake of water from the public network or bottled (litters of water per day) and habitual consumption of animal fats; and analgesic intake. Subjects were asked if they used an analgesic at least once a week for a month or more before the date of inclusion in the study (date of diagnosis of BC in the cases). Those who responded positively were asked about the number of weekly analgesic tablets and the condition for which the drug was prescribed.

Finally, both current and past occupational exposures and years of exposure to each of them were recorded. After a descriptive analysis of every occupational exposure, both in cases and controls, the assessment of professions as a BC risk factor was based on a meta-analysis of 263 articles [11], in which 61 occupations were classified following the codes of the International Standard Classification of Occupations (ISCO-58) [12]; the corresponding odds ratio (OR) for BC for every employment was then estimated: 42 occupations showed an increased incidence of BC, while 6 had a lower incidence. In the present study, and for analysis purposes, to summarize the different occupations collected along the participants’ working history, we assigned to every case and control the maximum risk of occupational exposure, expressed as the corresponding OR estimated in the above-mentioned meta-analysis [11].

This study was approved by the Ethics and Research Committee of the Hospital Universitari Mútua de Terrassa and conformed to the principles of the Declaration of Helsinki. All participants signed informed consent.

The statistical analysis of the data was carried out using the IBM SPSS version 26 program, including measures of central tendency and dispersion for the quantitative variables, and the frequencies with the corresponding percentages for the qualitative variables. For all the variables (risk factors) collected, the OR was initially obtained by means of a conditional logistic regression model, to account for matching, in which the dependent variable was a case or a control. The factors that showed statistical significance in the bivariate analysis were afterwards included in a multivariate conditional logistic regression model to obtain the corresponding adjusted OR and their 95% confidence intervals. In this model, the possible interaction between tobacco, coffee, and analgesic consumption was explored.

## 3. Results

A total of 306 participants were included in the study: 153 cases and 153 controls matched for age and sex. In each group, 89.5% (*n* = 137) of the participants were male. The mean age was 71.98 years (range 30–91; SD = 10.64) for the cases and 71.91 years (range 30–91; SD = 10.62) for the controls. The age distribution was identical in both groups, with 76.5% (*n* = 117) older than 65 years.

### 3.1. Family Background

There were no significant differences between cases and controls in terms of the presence of family history (68.8% vs. 31.3%, *p* = 0.123) (Table 1) nor globally (OR = 2.20; 95% CI 0.76–6.33) (Table 2).

### 3.2. Body Mass Index

No differences in body mass index (BMI) were observed between either group. Mean BMI was 28.62 ± 4.24 in cases and 28.04 ± 3.93 in controls (*p* = 0.21). The percentage of patients with obesity (BMI > 30) was similar between groups (30.7% in cases; 27.5% in controls; *p* = 0.615) (Table 1).

### 3.3. Smoking

On the one hand, blond tobacco was a risk factor for developing BC (OR 2.08; 95% CI 1.30–3.32; *p*= 0.002). The proportion of blond tobacco smokers was statistically different between cases and controls (52.94%, *n* = 81 vs. 34.64%, *n* = 53; *p* = 0.001). The differences remained statistically significant between groups when comparing the number of cigarettes per day (*p* = 0.002) and the years they have been smoking (*p* = 0.017). On the other hand, black tobacco was also a risk factor for developing BC (OR 2.67; 95% CI 1.55–4.58; *p* < 0.0001) (Table 1 and Table 2). Significant differences were found in the proportion of black tobacco smoking between cases and controls (44.44% (*n* = 68) vs. 24.84% (*n* = 38), *p* < 0.0001). However, no significant differences were observed in the number of daily cigarettes and the years of consumption of black tobacco in smokers of both groups.

### 3.4. Diet

The proportion of subjects consuming caffeinated or decaffeinated coffee, alcohol, tap or bottled water, and animal fats did not statistically differ between cases and controls. Controls had been consuming decaffeinated coffee for more years than cases (*p* = 0.007). Among the subjects who consumed alcohol, the patients with BC had a higher daily alcohol intake than those in the control group (*p* = 0.044), without any differences in the years of consumption. Overall, no variables related to diet seemed to behave as risk factors for developing BC (Table 1 and Table 2).

### 3.5. Analgesics Treatments

Eighty-six subjects were taking analgesics (seven tablets weekly in 73 subjects). There was a statistically significant difference in the consumption of analgesics between the cases (37.91%, *n* = 58) and the controls (18.30%, *n* = 28) (*p* < 0.0001) (Table 1). Analgesic consumption was a risk factor for developing BC (OR 2.67; 95% CI 1.55–4.58) (Table 2). None of the subjects had been prescribed pain-relieving drugs for BC.

To explore the possible interaction between tobacco, coffee consumption, and analgesics, first-order interaction terms were included in a conditional logistic regression model. The interaction between tobacco and coffee was not significant (neither for the global coffee variable, nor for the caffeinated coffee variable); in contrast, a significant interaction between tobacco and analgesics was pointed out (*p* = 0.012). Therefore, conditional odds ratios for coffee and analgesic consumption were estimated after stratifying by smoking (Table 3). In that way, analgesics showed a statistically significant association with bladder cancer in nonsmokers (OR 10.00; 95% CI 1.28–78.12; *p* = 0.028) but not in smokers (OR= 1.080; 95% CI 0.83–3.90; *p* = 0.136).

### 3.6. Occupational Exposure

When recording work history both in cases and controls, many different occupations were listed, leading to very small numbers in most of the professions (Table 4). Due to this high variability of working settings, the role of occupational exposure in relation to bladder cancer was assessed by using the previously defined variable, which assigned to each subject the estimated OR corresponding to the profession with the maximum estimated occupational risk for BC. Therefore, the occupational level of estimated risk of BC ranged in the study sample from 0.69 to 1.58, with percentiles 25, 50, and 75 being 1.10, 1.11, and 1.17, respectively. There was a statistically significant association between bladder cancer (being case or control) and the occupational level of risk for BC (OR = 6.72; 95% CI 1.06–42.7; *p* = 0.043). More than 75% of the cases and controls were retired at the time of inclusion in the study (76.47% in cases and 78.43 in controls, *p* = 0.682).

### 3.7. Multivariable Model

Finally, factors that have shown a statistically significant association with being a case of BC in the bivariate analysis were considered for inclusion in a multivariable conditional logistic regression model (Table 5). Adjusted by tobacco consumption and intake of analgesics, the occupational level of exposure was an independent risk factor for the development of BC (OR: 8.63, 95% CI 1.04–71.94, *p* = 0.046).

## 4. Discussion

The incidence of BC in Vallès Occidental remains one of the highest in men and one of the lowest in women, at a European and global level, despite the decline in tobacco consumption and industrial activity over the years [6]. Compared to data published during the period 1992–1994, both the crude annual incidence and age-adjusted incidence have increased in both sexes, although the increase in men is notably higher [5,13]. The high incidence of BC in men, some 25 years later, could be related to a high prevalence in this area of well-known risk factors such as smoking, residence in industrialized areas, and occupational exposure to certain carcinogenic products [13,14]. The analysis of possible environmental factors involved in this area showed that although the annual average concentrations of possible air and water pollutants were within the regulatory limit values, the maximum levels detected were usually higher than what was established [6]. In this case–control study, we try to identify other risk factors specifically related to BC in our health area.

The genetic involvement in bladder cancer is becoming increasingly well known. The risk of BC is twice as high in first-degree relatives of patients with BC, in relation to certain inherited genetic factors [7,15]. However, because of a lack of statistical power, the current results did not find significant differences between cases and controls in the presence of family history.

Smoking is also an important risk factor for BC in our area, both for dark tobacco and blond tobacco. Interestingly, both the number of daily cigarettes consumed and the years of smoking are significantly higher in the cases than in the controls for light tobacco, but not for dark tobacco. Although dark tobacco had traditionally been attributed a much higher risk than blond tobacco, the largest study to evaluate the effects of dark tobacco versus the use of blond tobacco on BC in Spain showed that the risk was only 40% higher for dark tobacco smokers compared to blond tobacco smokers, and this difference was not statistically significant [16]. In addition, smoking continues to be an important risk factor in our environment despite a significant decreased prevalence over time [6]. In Catalonia, in 1994, the prevalence of smokers in the population over 15 years of age was 42.3% in men and 20.7% in women. In 2018, the prevalence in men decreased to 30.9, although it remained high in the 35–44 age group (40.3%). In women, however, the 2018 prevalence of smoking was unchanged (20.5%), the highest figures being found between 25–34 years (31.6%). Previous studies showed that only 15.1% of the BC cases diagnosed during 1993–1995 in the Vallès Occidental had never smoked [13,17].

Various dietary factors have been investigated in numerous studies as likely risk factors for developing BC, with conflicting results. Although a higher fluid intake has been suggested to reduce the incidence of BC by diluting carcinogenic substances and promoting more frequent urination, thus limiting their effect on urothelial cells, studies focused on this hypothesis have not been able to validate it [18,19]. The chlorination of water, with the consequent level of trihalomethanes, has been considered an important carcinogenic risk of BC [20], and some studies have cited a higher BC risk in consumers of tap water due to the presence of trihalomethanes [21], or even as a risk factor independently of the chlorination [22]. Traditionally, coffee intake had been associated with a slight increase in the incidence of BC in smokers [22]. A recent meta-analysis of 10 cohorts and some case–control studies found no evidence of an association between BC and coffee intake [23]; nor has alcohol has been shown to be a risk factor for BC [24]. Finally, some studies have observed a relationship between the consumption of processed meat and animal proteins, and an increased risk of BC [25,26]. We have not found differences between cases and controls in any risk factor related to diet (consumption of coffee, alcohol, bottled or tap water, and animal fats). The fact that the subjects in the control group had been consuming decaffeinated coffee for more years could suggest a doubtful protective effect against BC.

Overweight and obesity have been described as risk factors in BC [27]. BMI has been associated with a linear rise in BC, with risk increasing by 4.2% for each increase of 5 mg/m^2^. However, this relationship may be biased by the fact that high BMIs are related to bad habits, such as little physical activity and inadequate diet [28]. In our study, we found no relationship between BC and BMI.

The association between the use of different types of analgesics and BC risk is controversial. On the one hand, there was strong experimental and epidemiological evidence that nonsteroidal anti-inflammatory drugs (NSAIDs) and cyclooxygenase 2 (COX-2) inhibitors might have a potential as cancer chemopreventive agents [29]. For example, ibuprofen, naproxen, indomethacin, piroxicam, and celecoxib inhibit BC development in a variety of human and animal models [29,30]. However, other studies describe an increased risk of BC associated with the use of phenacetin-containing analgesics, particularly with longer use. There are doubts about the association of paracetamol with BC, even though it is a metabolite of phenacetin [31,32]. Nor was regular use of any NSAID, including aspirin, associated with a statistically significant lower BC risk [31]. In a meta-analysis that included 17 articles on BC risk and analgesic use (8 cohort studies and 9 case–control studies), with a total of 10,618 cases of bladder cancer, there was no significant association between paracetamol use, aspirin, or other types of NSAIDs and BC risk [33]. However, NSAID use has been significantly associated with a 43% reduction in BC risk among nonsmokers but not among active smokers [15]. COX-2 expression is associated with increased tumor development. In smokers, both the expression and the activity of COX-2 are increased in urothelial tissues, but the anticancer effects of NSAIDs against COX-2 seem to be counteracted by the carcinogenic effect of smoking [34]. Our study did not show any protective effect of analgesics. In addition, the use of analgesics was related to BC in nonsmokers, thus being a risk factor independent of tobacco.

Occupational exposure to carcinogens such as aromatic amines (benzidine, 4-aminobiphenyl, 2-naphthylamine, 4-chloro-o-toluidine), polycyclic aromatic hydrocarbons, and chlorinated hydrocarbons, is considered the second most important risk factor for BC after smoking [9,11,15]. Approximately 20–25% of all BC are related to such exposure, mainly in industrial areas where paint, dyes, rubber, textiles, leather, metals, and petroleum products are processed, with a latency period of several decades [35]. Although in recent years the extent and pattern of occupational exposure have drastically changed due to an improved awareness of occupational safety measures [9], some occupations, such as those in the chemical sector, are still considered as risk factors; rubber, textile, printing, and other industries are probably linked to exposure to carcinogenic agents. The relationship of BC with hair dye, or even with the hairdressing professional who handles such products, is still controversial [9].

In our environment, our data pointed to profession as a risk factor for developing BC, independently of tobacco and analgesics consumption. A meta-analysis of 263 publications [11] concluded that although there is evidence of a decrease in the incidence and occupational mortality of BC, certain occupations are still associated with a high incidence or greater risk of mortality from BC: there is an increase in the incidence of BC in 42 of 61 occupations analyzed and of BC-specific mortality in 16 of 40, although not all studies had explored specific mortality. The highest combined incidence risks are seen in tobacco workers (RR 1.72, 95% CI 1.37–2.15) and dye workers (RR 13.4, 95% CI 1.5–48.2). However, the highest RR reported in any study was for factory workers overall (RR 16.6; 95% CI, 2.1–131.3). In terms of grouped disease-specific mortality, it is higher for metal workers (RR, 10.2; 95% CI, 6.89–15.09) and gardeners (RR, 5.5; 95% CI, 0.84–35.89) with the highest disease-specific mortality reported in any study for chemical workers (RR, 27.1; 95% CI, 11.7–53.4) [11]. These high BC incidences and mortality persist despite improvements in workplace safety measures, and efforts to reduce the impact of BC on workers should be directed at the highest-risk occupations.

The textile industry constituted the economic base of the Vallès Occidental region from the mid-19th century to the 1970s. This fact justified the performance of BC incidence and population-based case–control studies whose objectives were to assess occupational risk factors for BC in this area [5,13,17]. These studies demonstrated that tobacco consumption was strongly associated with BC [13]. However, when analyzing BC risk associated with exposure in the textile industry as part of a large case–control study carried out in five areas of Spain (Asturias, Alicante, Barcelona, Tenerife and Vallès/Bages), working in the textile industry was not associated with a higher BC risk. However, specific occupations within the textile industry (for example, weavers) and specific locations (winding, warping and gluing, and weaving room), as well as having contact with specific materials (synthetics and cotton), showed an increased BC risk [14].

A limitation of this case–control study is the small sample size, which does not allow us to make comparisons between men and women in terms of tobacco consumption, use of analgesics, and professions. It would also have been interesting to have information on the type of analgesics being consumed, to assess the differences between steroidal drugs and NSAIDs.

## 5. Conclusions

We conclude that consumption of black and blond tobacco, the use of analgesics in nonsmoking patients, and profession are independent risk factors for the development of BC in our environment. The decline in smoking in the population, especially in men, and the improvements in job security have not been sufficient to reduce this high incidence of BC.

## Figures and Tables

**Table 1 jcm-12-00728-t001:** Bivariate assessment of risk factors involved in bladder cancer in West Vallès Occidental.

Variable		Cases	Controls	*p*-Value ^1^
Family background of bladder cancer	*n* (%)	11 (68.8)	5 (31.3)	0.123
Obesity (BMI > 30)	*n* (%)	47 (30,7)	42 (27.5)	0.615
Blond tobacco	*n* (%) Number cigarettes/day. mean (SD) Years, mean (SD)	81 (52.94) 21.64 (12.10) 35.95 (13.69)	53 (34.64) 15.64 (8.77) 29.74 (15.73)	**0.001** **0.002** **0.017**
Black tobacco	*n* (%) Number cigarettes/day. mean (SD) Years, mean(SD)	68 (44.44) 21.99 (14.81) 36.13 (16.24)	38 (24.84) 18.37 (11.71) 32.13 (14.67)	**<0.0001** 0.198 0.211
Caffeinated coffee	*n* (%) Number cups of coffee/day. mean (SD) Years, mean (SD)	122 (79.74) 2.04 (1.58) 47.84 (20.78)	111 (72.55) 2.05 (1.33) 45.92 (11.82)	0.141 0.946 0.392
Decaffeinated coffee	*n* (%) Number cups of coffee/day mean (SD) Years, mean (SD)	24 (15.69) 2.33 (1.71) 32.67 (18.67)	24 (15.69) 1.54 (0.93) 45.79 (12.86)	1.000 0.052 **0.007**
Alcohol consumption	*n* (%) Grams of alcohol/day. mean (SD) Years. Mean (SD)	101 (66.01) 38.22 (26.24) 47.90 (10.87)	98 (64.05) 31.33 (21.44) 47.61 (9.56)	0.720 **0.044** 0.843
Tap water	*n* (%) Liters/day, mean (SD)	53 (34.64) 1.82 (2.07)	47 (30.72) 1.28 (0.51)	0.465 0.086
Bottled water	*n* (%) Liters/day, mean (SD)	101 (66.01) 2.02 (3.09)	107 (69.93) 1.27 (0.53)	0.463 **0.013**
Animal fats	*n* (%) Quantity, mean (SD)	141 (92.16) 2.96 (1.64)	146 (95.42) 3.09 (1.70)	0.237 0.52
Analgesics treatments	*n* (%) Tablets/week, mean (SD)	58 (37.91) 6.83 (2.33)	28 (18.30) 6.36 (1.64)	**<0.0001** 0.34

*n* = number of individuals; SD: standard deviation. ^1^ Student *t* test for parametric variables and Mann–Whitney U for nonparametric variables, significance level <0.05. **In bold,** significant *p*-values.

**Table 2 jcm-12-00728-t002:** Odds ratio and 95% CI for the risk factors collected.

Variable	Conditional Logistic Regression	
	OR (95% CI)	*p*
Age	1.16 (0.83–1.63)	0.393
Family background of bladder cancer	2.20 (0.76–6.33)	0.144
Body mass index (BMI)	1.03 (0.98–1.09)	0.217
Obesity (BMI > 30)	1.17 (0.71–1.92)	0.529
Blond tobacco	**2.08 (1.30–3.32)**	**0.002**
Black tobacco	**2.67 (1.55–4.58)**	**<0.0001**
Caffeinated coffee	1.61 (0.89–2.90)	0.112
Decaffeinated coffee	1.00 (0.51–1.96)	1.000
Alcohol consumption	1.11 (0.66–1.87)	0.691
Tap water	1.18 (0.74–1.88)	0.480
Bottled water	0.85 (0.53–1.34)	0.480
Animal fats	1.83 (0.68–4.96)	0.232
Analgesics treatments	**2.67 (1.55–4.58)**	**<0.0001**

OR = odds ratio; 95% CI = 95% confidence interval for OR. **In bold,** significant *p*-values OR (significance level < 0.05).

**Table 3 jcm-12-00728-t003:** Estimated odds ratios for coffee and analgesics consumption as risk factors for bladder cancer, stratified by smoking status.

Variable	Cases (*n* = 152) *n* (No/Yes)	Controls (*n* = 152) *n* (No/Yes)	Nonsmoker		Smoker	
			OR (95% CI)	*p*-Value ^1^	OR (95% CI)	*p*-Value ^1^
Coffee consumption	17/136	24/129	1.00 (0.14–7.10)	1.000	0.43 (0.11–1.66)	0.220
Caffeinated coffee consumption	31/122	42/111	1.50 (0.25–8.98)	0.657	0.82 (0.34–1.97)	0.655
Analgesics treatment	95/58	125/28	**10.00 (1.28–78.12)**	**0.028**	1.80 (0.83–3.90)	0.136

Odds ratios (OR) and 95% confidence intervals (95% CI) were estimated by means of conditional logistic regression. **In bold,** *p*-values and significant OR (^1^ significance level < 0.05).

**Table 4 jcm-12-00728-t004:** Comparison between cases and controls of the different types of occupation and the years spent.

Occupational		Cases	Controls	*p*-Value ^1^
Homemaker	*n* (%) Mean years (SD)	7 49.14 (13.09)	1 60	**0.032** 0.467
Textile	*n* (%) Mean years (SD)	30 24.70 (16.79)	21 27.95 (18.14)	0.167 0.513
Mechanic	*n* (%) Mean years (SD)	17 26.65 (19.93)	9 20.78 (14.90)	0.102 0.447
Truck driver	*n* (%) Mean years (SD)	9 20.67 (13.53)	12 28.92 (14.38)	0.652 0.198
Painter	*n* (%) Mean years (SD)	9 26.89 (19.20)	5 20.40 (13.76)	0.275 0.520
Rubber-plastic	*n* (%) Mean years (SD)	3 9.67 (8.39)	1 39.00	0.623 0.094
Asbestos	*n* (%) Mean years (SD)	2 9.00 (9.90)	0 -	0.498 -
Printing	*n* (%) Mean years (SD)	5 36.00 (20.16)	3 40.00 (12.77)	0.723 0.772
Agriculture	*n* (%) Mean years (SD)	17 13.29 (10.60)	8 8.75 (2.05)	0.093 0.247
Laundry	*n* (%) Mean years (SD)	1 6.00	- -	**-** **-**
Building	*n* (%) Mean years (SD)	30 31.17 (17.88)	28 24.36 (17.87)	0.884 0.153
Welder	*n* (%) Mean years (SD)	2 46.00 (5.66)	2 30.00 (21.21)	1.00 0.411
Hairdressing	*n* (%) Mean years (SD))	1 8.00	3 35.67 (23.12)	0.315 0.409
Dyes	*n* (%) Mean years (SD)	2 3.50 (0.70)	0	0.498 -
Metallurgy	*n* (%) Mean years (SD)	17 21.82 (12.04)	24 26.67 (19.76)	0.314 0.375
Chemistry	*n* (%) Mean years (SD)	7 25.71 (13.99)	6 34.67 (16.70)	0.777 0.315
Mining	*n* (%) Mean years (SD)	1 2.00	0	1.00
Fire-fighter	*n* (%) Mean years (SD)	0	1 35.00	1.00 -
Electricity	*n* (%) Mean years (SD))	5 33.00 (17.19)	53 1.80 (14.38)	1.00 0.908
Feeding	*n* (%) Mean years (SD)	4 12.75 (15.15)	8 22.62 (15.24)	0.378 0.314
Sales	*n* (%) Mean years (SD))	20 29.35 (15.94)	22 30.14 (16.27)	0.740 0.875
Waiter	*n* (%) Mean years (SD)	7 19.57 (15.10)	7 16.00 (15.71)	1.00 0.672
Health area	*n* (%) Mean years (SD)	5 35.60 (11.54)	5 20 (11.25)	1.00 0.062
Office	*n* (%) Mean years (SD)	18 35.61 (14.62)	21 38.24 (13.84)	0.732 0.568
Teaching	*n* (%) Mean years (SD)	1 4.00	10 36.40 (5.08)	**0.010** **<0.0001**
Others	*n* (%) Mean years (SD)	29 29.03 (14.40)	32 32.66 (15.67)	0.775 0.353

*n* = number of individuals; ^1^
**In bold,** *p*-value and significant OR (significance level <0.05).

**Table 5 jcm-12-00728-t005:** Independent predictive factors of bladder cancer. Conditional logistic regression model.

	B	*p*-Value ^1^	OR	95.0% CI for OR	
				Lower	Upper
Occupational risk	2.156	**0.046**	8.634	1.036	71.941
Blond tobacco	1.272	**<0.0001**	3.567	1.917	6.639
Dark tobacco	1.448	**<0.0001**	4.255	2.178	8.311
Analgesics	0.912	**0.004**	2.490	1.336	4.643

^1^**In bold,** *p*-values and significant OR (significance level <0.05).

## Data Availability

Data sharing is not applicable to this article as no datasets were generated or analyzed during the current study.

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
