# Peer review of "Risk Factors Involved in the High Incidence of Bladder Cancer in an Industrialized Area in North-Eastern Spain: A Case–Control Study"

_jcm, 2023, doi:10.3390/jcm12020728_

Round 1

Reviewer 1 Report

Thank you for giving me the opportunity to review this article. The authors had conducted previous research on environmental factors for bladder cancer in northeastern Spain (Ref. 5). The present study was a case-control study in which a control group was created to examine the influence of occupational factors as well. Because of the large variety of occupations, logistic regression analysis was conducted using estimated risk for each occupation. The results of the multivariate analysis showed that an occupational factor could be an independent risk factor for incidences of bladder cancer, as well as smoking and analgesics treatment. Although the number of cases is small, this is a unique study focusing on occupational factors; therefore, this paper is considered to be worth publishing.   The followings are my comments and questions. (Abstract) -306 participants were included---(page 1, line 20). Numbers at the beginning of a sentence should be written in letters. (Introduction) The authors mentioned the regionality of bladder cancer. However, cancer development by environmental factors is not considered to be a risk only for bladder cancer. Please describe other cancers' medical situations in this region. (Materials and Methods) -What does medical family history mean? Does it include only bladder cancer or other cancers? (Table 4) -Housewife and Fireman seem to have gendered meanings. According to ISCO-58, "Homemaker" and "Fire-fighter" would be better. (Analgesics treatment) Associations between analgesics and cancer development have been controversial. Does "Analgesic treatments" in this study mean analgesic usage before a diagnosis of bladder cancer? If analgesics were used for cancer-related pain, that would be a confounding bias.

Reviewer 2 Report

Caballero et al. have demonstrated the risk factors involved in the development of bladder cancer in an industrialized area in north-eastern Spain. The concept of this article is interesting to read, but the findings we can speculate is strongly limited.

 Tobacco smoking is one of the major risk factors for UC development, and are already wildly known all over the world. Analgesics use is still in the middle of the discussion whether these drugs truely raise the risks of UC development since numerous patients would use them in daily pracitce.  The most novel findings of this study is that they classified the risk of UC development per occupations, but the case and control numbers of patients are too small to lead to a concrete  conclusion. Thus, I consider that the current study is yet mature for further publication. 

Our advice is to increase numbers of UC patients and controls to reduce the entire selection bias.

Round 2

Reviewer 2 Report

Thank you for providing the detailed response to our questions. As the authors mentioned, we agree that reliability and validity are different concept, and understand that the authors seeked to minimize the selection bias by chosing methods for patient seleciton and also by using matched cohorts. 

Our concerns have been solved by their detailed comments.